# DMT alters cortical travelling waves

**Andrea Alamia[1‡]\*, Christopher Timmermann[2,3‡]\*, David J Nutt[3], Rufin VanRullen[1,4†], Robin L Carhart-Harris[3†]**

[1]Cerco, CNRS Université de Toulouse, Toulouse, France; [2]Computational, Cognitive and Clinical Neuroscience Laboratory (C3NL), Faculty of Medicine, Imperial College, London, United Kingdom; [3]Centre for Psychedelic Research, Division of Psychiatry, Department of Brain Sciences, Imperial College London, London, United Kingdom; [4]Artificial and Natural Intelligence Toulouse Institute (ANITI), Toulouse, France

**Abstract** Psychedelic drugs are potent modulators of conscious states and therefore powerful tools for investigating their neurobiology. N,N, Dimethyltryptamine (DMT) can rapidly induce an extremely immersive state of consciousness characterized by vivid and elaborate visual imagery. Here, we investigated the electrophysiological correlates of the DMT-induced altered state from a pool of participants receiving DMT and (separately) placebo (saline) while instructed to keep their eyes closed. Consistent with our hypotheses, results revealed a spatio-temporal pattern of cortical activation (i.e. travelling waves) similar to that elicited by visual stimulation. Moreover, the typical top-down alpha-band rhythms of closed-eyes rest were significantly decreased, while the bottom-up forward wave was significantly increased. These results support a recent model proposing that psychedelics reduce the 'precision-weighting of priors', thus altering the balance of top-down versus bottom-up information passing. The robust hypothesis-confirming nature of these findings imply the discovery of an important mechanistic principle underpinning psychedelic-induced altered states.

**\*For correspondence:**
andrea.alamia@cnrs.fr (AA);
c.timmermann-slater15@imperial.
ac.uk (CT)

[†]These authors contributed
equally to this work
[‡]These authors also contributed
equally to this work

**Competing interests:** The
authors declare that no
competing interests exist.

**Reviewing editor:** Virginie van
Wassenhove, CEA, DRF/I2BM,
NeuroSpin; INSERM, U992,
Cognitive Neuroimaging Unit,
France

## Introduction

N,N, Dimethyltryptamine (DMT) is a mixed serotonin receptor agonist that occurs endogenously in several organisms (*Christian et al., 1977*; *Nichols, 2016*) including humans (*Smythies et al., 1979*), albeit in trace concentrations. DMT, which is a classic psychedelic drug, is also taken exogenously by humans to alter the quality of their consciousness. For example, synthesized compound is smoked or injected but it has also been used more traditionally in ceremonial contexts (e.g. in Amerindian rituals). When ingested orally, DMT is metabolized in the gastrointestinal (GI) system before reaching the brain. Its consumption has most traditionally occurred via drinking 'Ayahuasca', a brew composed of plant-based DMT and $\beta$ –carbolines (monoamine oxidize inhibitors), which inhibit the GI breakdown of the DMT (*Buckholtz and Boggan, 1977*). Modern scientific research has mostly focused on intravenously injected DMT. Administered in this way, DMT's subjective effects have a rapid onset, reaching peak intensity after about 2–5 min and subsiding thereafter, with negligible effects felt after about 30 min (*Strassman, 2001*; *Strassman, 1995a*; *Timmermann et al., 2019*).

Previous electrophysiological studies investigating changes in spontaneous (resting state) brain function elicited by ayahuasca have reported consistent broadband decreases in oscillatory power (*Riba et al., 2002*; *Timmermann et al., 2019*), while others have noted that the most marked decreases occur in α-band oscillations (8–12 Hz) (*Schenberg et al., 2015*). Alpha decreases correlated inversely with the intensity of ayahuasca-induced visual hallucinations (*Valle et al., 2016*) and are arguably the most reliable neurophysiological signature of the psychedelic state identified to-date (*Muthukumaraswamy et al., 2013*) – with increased signal diversity or entropy being another particularly reliable biomarker (*Schartner et al., 2017*). In the first EEG study of the effects of pure DMT on on-going brain activity, marked decrease in the α and β (13–30 Hz) band power was

observed as well as increase in signal diversity (*Timmermann et al., 2019*). Increase in lower frequency band power (δ = 0.5–4 Hz and θ = 4–7 Hz) also became evident when the signal was decomposed into its oscillatory component. Decreased alpha power and increased signal diversity correlated most strongly with DMT's subjective effects – consolidating the view that these are principal signatures of the DMT state, if not the psychedelic state more broadly.

Focusing attention onto normal brain function, outside of the context of psychoactive drugs, electrophysiological recordings in cortical regions reveal distinct spatio-temporal dynamics during visual perception, which differ considerably from those observed during closed-eyes restfulness. It is possible to describe these dynamics as oscillatory 'travelling waves', i.e. fronts of rhythmic activity which propagate across regions in the cortical visual hierarchy (*Lozano-Soldevilla and VanRullen, 2019*; *Muller et al., 2014*; *Sato et al., 2012*). Recent results showed that travelling waves can spread from occipital to frontal regions during visual perception, reflecting the forward bottom-up flow of information from lower to higher regions. Conversely, top-down propagation from higher to lower regions appears to predominate during quiet restfulness (*Alamia and VanRullen, 2019*; *Halgren et al., 2019*; *Pang et al., 2020*).

Taken together these results compel us to ask how travelling waves may be affected by DMT, particularly given their association with predictive coding (*Alamia and VanRullen, 2019*; *Friston, 2019*) and a recent predictive coding inspired hypothesis on the action of psychedelics ('REBUS') – which posits decreased top-down processing and increased bottom-up signal passing under these compounds (*Carhart-Harris and Friston, 2019*). Moreover, DMT lends itself particularly well to the testing of this hypothesis as its visual effects are so pronounced. Given that visual perception is associated with an increasing in forward travelling waves and eyes-closed visual imagery under DMT can feel as if one is 'seeing with eyes shut' (*de Araujo et al., 2012*) – does a consistent increase in forward travelling waves under DMT account for this phenomenon?

Here we sought to address these questions by quantifying the amount and direction of travelling waves in a sample of healthy participants who received DMT intravenously, during eyes-closed conditions. We hypothesized that DMT acts by disrupting the normal physiological balance between top-down and bottom-up information flow, in favour of the latter (*Carhart-Harris and Friston, 2019*). Moreover, we ask: does this effect correlate with the vivid 'visionary' component of the DMT experience? Providing evidence in favour of this hypothesis would indicate that forward travelling waves do play a crucial role in conscious visual experience, irrespective of the presence of actual photic stimulation.

## Results

### Quantifying travelling waves

As demonstrated by both theoretical and experimental evidence (*Nunez, 2000*; *Nunez and Srinivasan, 2014*; *Nunez and Srinivasan, 2009*), in most systems, including the human brain, travelling waves occur in groups (or packets) over some range of spatial wavelengths having multiple spatial and temporal frequencies. Given any configurations of electrodes, only parts of these packets can be successfully detected, i.e. waves shorter than the spatial extent of the array, and waves longer than twice the electrode separation distance (Nyquist criterion in space). In scalp recordings, the shorter waves may be mostly removed by volume conduction. As a consequence, waves recorded directly from the cortex emphasize shorter waves than the scalp recorded waves. Specifically, in the case of small cortical arrays, the overlap between cortical and scalp data may be minimal, and the estimated wave properties (including propagation direction) may differ. Additionally, it is important to consider that when waves are travelling in multiple directions at nearly the same time in 'closed' systems (e.g. the cortical/white matter), waves either damp out or interfere with each other to form standing waves (e.g. alpha waves travelling both forward and backward). It is reasonable to assume that the behaviour of these properties will relate to global brain and mind states, and be sensitive to state-altering psychoactive drugs (*Nunez, 2000*; *Nunez and Srinivasan, 2014*; *Nunez and Srinivasan, 2009*).

Practically, we measure the waves' amount and direction with a method devised in our previous studies (*Alamia and VanRullen, 2019*; *Pang et al., 2020*). We slide a one-second time-window over the EEG signals (with 0.5 s overlap). For each time-window, we generate a 2D map (time/electrodes)

by stacking the signals from five central mid-line electrodes (Oz to FCz, see **Figure 1**). For each map, we then compute a 2D-FFT, in which the upper- and lower-left quadrant represent the power of forward (FW) and backward (BW) travelling waves, respectively (since the 2D-FFT is symmetrical around the origin, the lower- and upper-right quadrants contain the same information). From both quadrants we extracted the maximum values, representing the raw amount of FW and BW waves in that time-window. Next, we performed the same procedure after having shuffled the electrodes' order, thereby disrupting spatial information (including the waves' directionality) while retaining the same overall spectral power. In other words, the surrogate measures reflect the amount of waves expected solely due to the temporal fluctuations of the signal. After having computed the maximum values for the FW and BW waves of the surrogate 2D-FFT spectra one hundred times (and averaging the 100 values), we compute the net amount of FW and BW waves in decibel (dB), by applying the following formula:

$$WdB = 10 * \log_{10} \frac{W}{Wss}$$

where W represents the maximum value extracted for each quadrant (i.e. forward FW or backward BW), and Wss the respective surrogate value. Importantly, this value – expressed in decibel – represents the net amount of waves against the null distribution. In other words, it is informative to

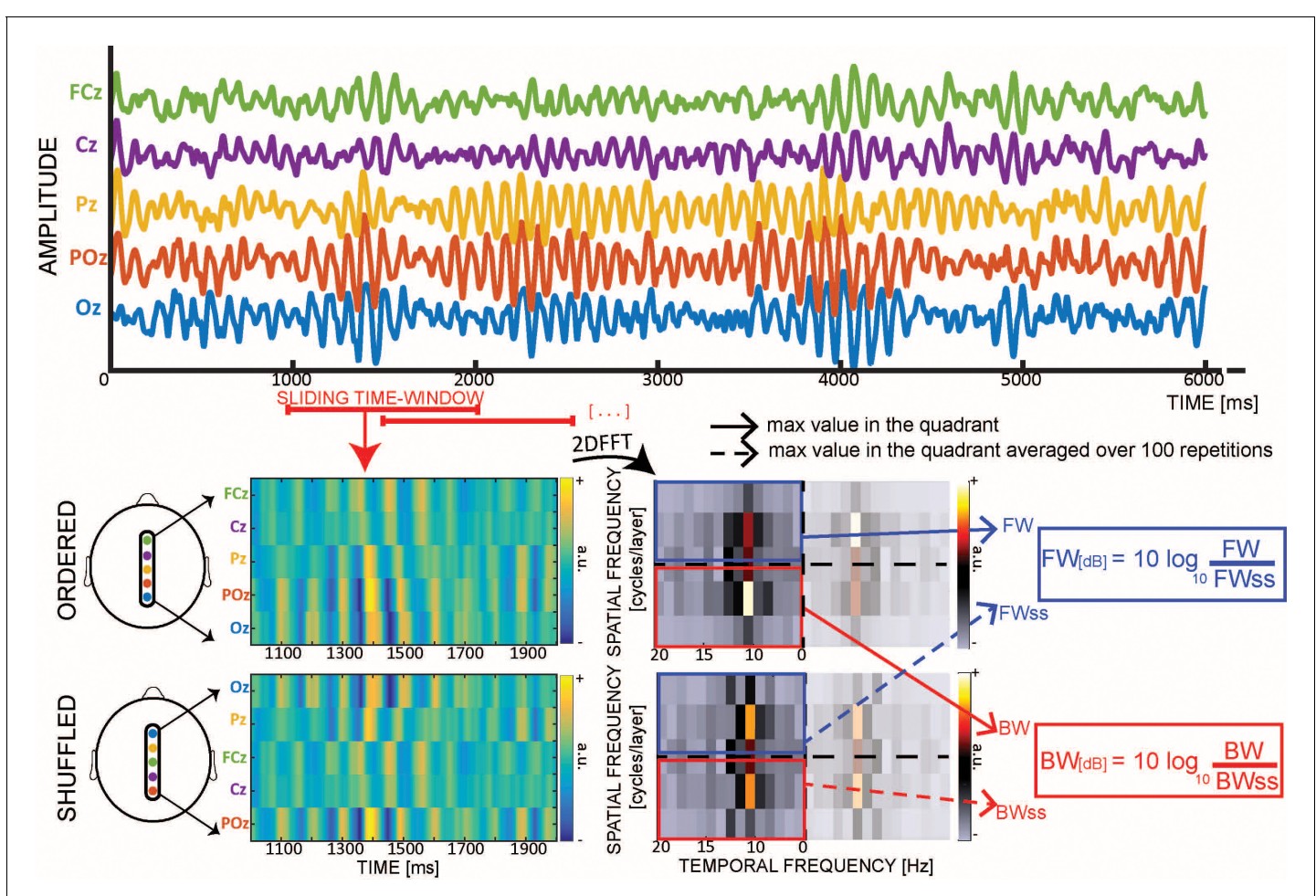

**Figure 1.** Quantifying cortical waves. From each 1 s EEG epoch we extract a 2D-map, obtained by stacking signals from five midline electrodes. For each map we compute a 2D-FFT in which the maximum values in the upper- and lower-left quadrants represent respectively the amount of forward (FW – in blue) and backward (BW – in red) waves. For each map, we also compute surrogate values by shuffling the electrodes' order 100 times, so as to retain temporal fluctuations while disrupting the spatial structure of the signals (including any travelling waves). Eventually, we compute the wave strength in decibel (dB) by combining the real and the surrogate values.

compare this value to zero, to assess the significance of waves. On the other hand, a direct comparison between FW and BW waves in each time-bin is not readily interpretable, as it is possible to simultaneously record waves propagating in both directions—as observed during visual stimulation epochs (see below). In addition, it's important to note that our waves' analysis focuses on the sensor level, as source projections presents a number of important limitations, such as impairing long-range connections, as well as smearing of signals due to scalp interference (*Alexander et al., 2019*; *Freeman and Barrie, 2000*; *Nunez, 1974*).

## Does DMT influence travelling waves?

After defining our measure of the waves' amount and direction, we investigated whether the intake of DMT alters the cortical pattern of travelling waves. Participants underwent two sessions in which they were injected with either placebo or DMT (see Materials and methods for details). Importantly, during all of the experiments, participants rested in a semi-supine position, with their eyes closed. EEG recordings were collected 5 min prior to drug administration and up to 20 min after. The left column of *Figure 2A* shows the amount of BW and FW waves in the 5 min preceding and following drug injection (either placebo or DMT). Consistent with previous observations on independent data (*Alamia and VanRullen, 2019*), during quiet closed-eyes restfulness a significant amount of BW waves spread from higher to lower regions (as confirmed by a Bayesian t-test against zero for both DMT and Placebo conditions, $BFs_{10} >> 100$, error <0.01%, 95% Credible Intervals (CI) DMT: [0.221, 0.637], Placebo: [0.273, 0.666]), whereas no significant waves propagate in the opposite FW direction (Bayesian t-test against zero: $BFs_{10} < 0.15$, error <0.01%; 95% CI DMT: [−0.424, 0.088], Placebo: [−0.372, 0.110]). However, after DMT injection, the cortical pattern changed drastically: the amount of BW waves decreased (but remaining significantly above zero – $BFs_{10} = 12.6$, 95% CI: [0.057, 0.322]), whereas the amount of FW waves increased significantly above zero ($BF_{10} = 5.4$ 95% CI: [0.027, 0.336]). These results, obtained by comparing the amount of waves before and after injection (pre-post factor) of Placebo or DMT (drug factor), were confirmed by two Bayesian ANOVA performed separately on BW and FW waves (all factors including interactions reported $BFs_{10} >> 100$, error <2%), and were not confounded by differences in dosage (see Materials and methods and *Figure 2—figure supplement 1*). A power analysis comparing DMT and Placebo conditions after infusion for both FW and BW direction revealed values above 90% (FW case: $\mu_{DMT}=0.19$, $\mu_{PLACEBO} = -0.20$ and $\sigma = 0.29$ yields to power equals to 0.9168; BW case: $\mu_{DMT} = 0.18$, $\mu_{PLACEBO} = 0.51$ and $\sigma = 0.25$ gives power equals to 0.9205; in both cases, we considered a type I error rate of 5%).

In order to explore different propagation axes than the midline, we ran the same analysis on one array of electrodes running from posterior right to anterior left regions, and one from posterior left to anterior right ones: in both cases we obtained similar results as for the midline electrodes, i.e. an increase and a decrease of FW and BW waves, respectively, following DMT infusion (see *Figure 2—figure supplement 2*). This suggests that the dominant natural propagation spread of travelling waves is along the axis that connects the furthest posterior and frontal recording channels. As a control, we additionally demonstrated that waves propagating from leftward to rightward regions (and vice versa), were not affected by DMT (see *Figure 2—figure supplement 2*). Besides, in-line with previous work on travelling waves (*Alexander et al., 2013*; *Alexander et al., 2006*), an additional analysis based on relative phases of the alpha band-pass signals over all channels, confirmed the same results, with DMT indeed disrupting the typical top-down propagation of alpha-band waves. Furthermore, we ran a more temporally precise analysis, on a minute-by-minute scale, testing the amount of FW and BW waves in the two conditions, as shown in the right panels of *Figure 2A*. in-line with previous studies (*Strassman, 1995a*; *Strassman, 1994*; *Timmermann et al., 2019*), the changes in cortical dynamics appeared rapidly after intravenous DMT injection, and began to fade after about 10 min. Confirming our previous analysis, we observed an increase in FW waves (asterisks in the upper-right panel of *Figure 2A* show FDR-corrected significant p-values when testing against zero) and a decrease in BW waves, which, nonetheless, remained above zero (all FDR-corrected p-values<0.05). To our initial surprise, the dynamics elicited by DMT injection were remarkably reminiscent of those observed in another study, in which healthy participants alternated visual stimulation with periods of blank screen, without any drug manipulation (*Pang et al., 2020*). Although a direct comparison is not statistically possible (because the two studies involved distinct subject groups and different EEG recording setups), we indirectly investigated the similarities between these two scenarios.

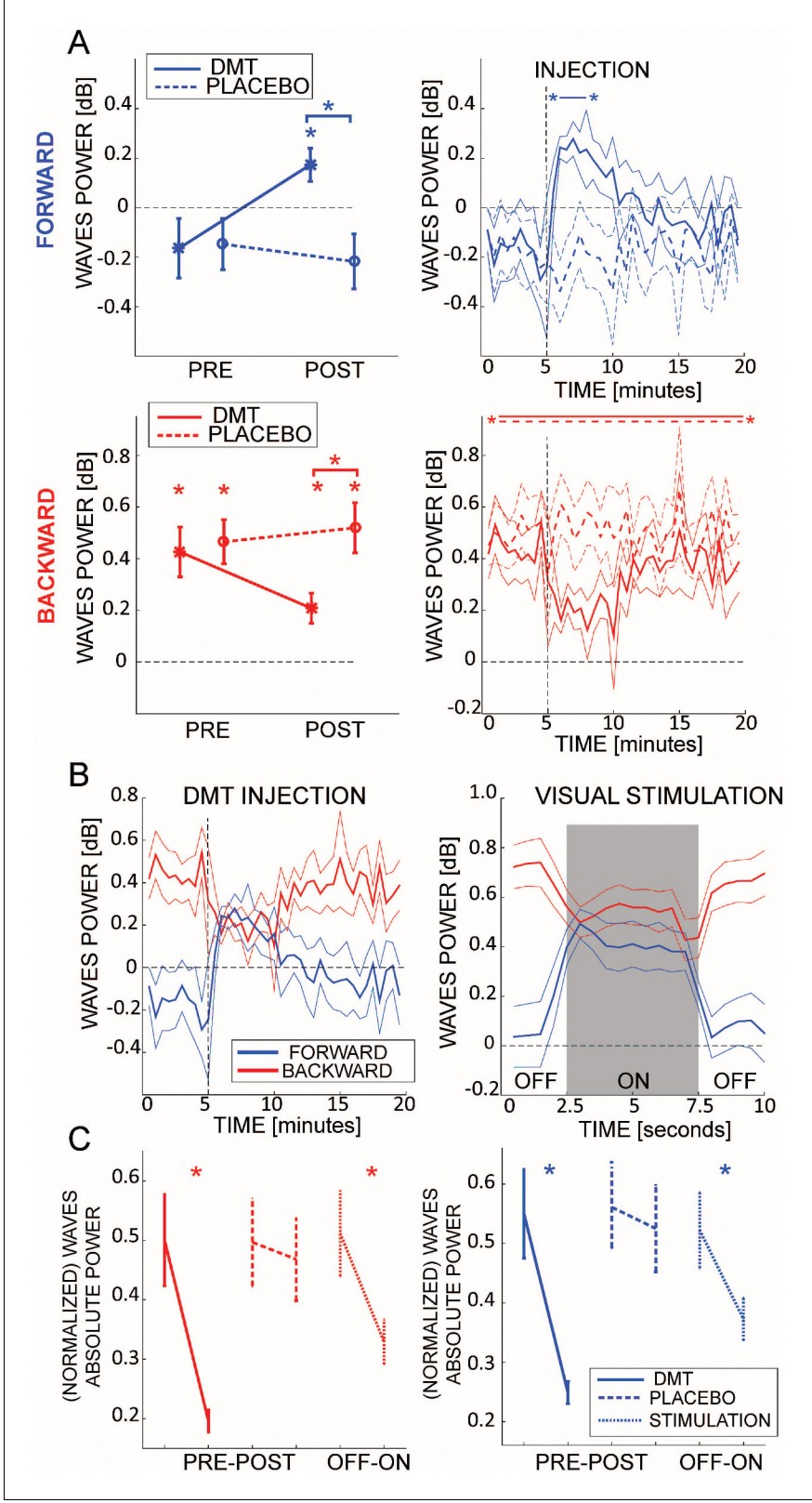

**Figure 2.** DMT influences cortical travelling waves. (**A**) In the left panels the net amount of FW (blue, upper panel) and BW (red, lower panel) waves is represented pre- and post-DMT infusion. While BW waves are always present, FW waves only rise significantly above zero after DMT injection, despite participants having closed eyes. Asterisks denote values significantly different than zero, or between conditions. The panels to the right describe the minute-by-minute changes in the net amount of waves. Asterisks denote FDR-corrected p-values for amount of waves significantly different than zero.

*Figure 2 continued on next page*

*Figure 2 continued*

(B) Comparison between the waves' temporal evolution after DMT injection (left panel) and with or without visual stimulation (right panel, from a different experiment in which participants, with open eyes, either watched a visual stimulus or a blank screen *Pang et al., 2020*). Remarkably, the waves' temporal profiles are very similar in the two conditions, for both FW and BW. (C) Comparison between changes in absolute power (as extracted from the 2D-FFT, that is FW and BW in *Figure 1*) due to DMT, placebo and visual stimulation. Remarkably, true photic visual stimulation and eyes-closed DMT induce comparably large reductions in absolute power. In fact, the effect with DMT appears to be even more pronounced (formal contrast not appropriate). Note that in the previous panels the changes in the net amount of waves were reported in dB, and occurred irrespective of the global power changes measured in panel C.

The online version of this article includes the following figure supplement(s) for figure 2:

**Figure supplement 1.** Changes in the amount of FW/BW waves as a function of dosage.
**Figure supplement 2.** FW and BW waves' direction along different axes.

## Comparison with perceptual stimulation

We recently showed that FW travelling waves increase during visual stimulation, whereas BW waves decrease, in-line with their putative functional role in information transmission (*Pang et al., 2020*). In *Figure 2B*, for the sake of comparison, we contrast the cortical dynamics induced by DMT (left panel) with the results of our previous study (right panel *Pang et al., 2020*), in which participants perceived a visual stimulus (label 'ON') or stared at a dark screen (label 'OFF'). Remarkably, mutatis mutandis, both FW and BW waves share a similar profile across the two conditions, increasing and decreasing respectively following DMT injection or visual stimulation. If we consider the absolute (maximum) power values derived from the 2D-FFT of each map (i.e. before estimating the surrogates and the waves' net amount in decibel) as an estimate of spectral power, we can read the results reported in *Figure 2C* as an overall decrease in oscillatory power following DMT injection, more specifically in the frequency band with the highest power values (i.e. alpha band, but see next paragraph) (*Muthukumaraswamy et al., 2013*; *Riba et al., 2002*; *Schenberg et al., 2015*; *Timmermann et al., 2019*). Such decrease in oscillatory power is also matched by a similar decrease induced by visual stimulation (all Bayesian t-test $BFs_{10} >> 100$). These results demonstrate that, despite participants having their eyes-closed throughout, DMT produces spatio-temporal dynamics similar to those elicited by true visual stimulation. These results therefore shed light on the neural mechanisms involved in DMT-induced visionary phenomena.

## Does DMT influence the frequency of travelling waves?

Previous studies showed that DMT alters specific frequency bands (e.g. alpha-band *Schenberg et al., 2015*), mostly by decreasing overall oscillatory power (*Riba et al., 2002*; *Timmermann et al., 2019*). Here, we investigated whether DMT influences not only the waves' direction but also their frequency spectrum. We compared the frequencies of the maximum peaks extracted from the 2D-FFT (see *Figure 1*) before and after DMT or Placebo injection. Before infusion, both FW and BW waves had a strong alpha-range oscillatory rhythm (*Figure 3A*, labeled 'PRE'). Remarkably, following DMT injection, the waves' spectrum changed drastically, with a significant reduction in the alpha-band, coupled with an increase in the delta and theta bands, for both FW (δ-band: $BF_{10} = 391.16$, θ-band: $BF_{10} = 19.23$, α-band: $BF_{10} = 16.04$, β-band: $BF_{10} = 0.64$; all errors < 0.001%) and BW waves (δ-band: $BF_{10} = 82.56$, θ-band: $BF_{10} = 30.58$, α-band: $BF_{10} = 549.54$, β-band: $BF_{10} = 1.43$; all errors < 0.005%). This result corroborates a previous analysis performed on EEG recordings from the same dataset (*Timmermann et al., 2019*) as well as independent data pertaining to O-Phosphoryl-4-hydroxy-N,N-DMT (psilocybin), a related compound (*Muthukumaraswamy et al., 2013*). Moreover, we investigated how DMT influences the amount of waves at each frequency.

As shown in *Figure 3B*, and in agreement with previous analyses, DMT induces an overall reduction in the amount of waves at each frequency, specifically in the alpha-band BW waves, but with the notable exception in the FW alpha band, in which DMT induces an increase in the waves' direction.

## What's the relationship between FW and BW waves?

From the left panel of *Figure 2B*, it seems that during the first minutes after DMT injection, both FW and BW waves are simultaneously present in the brain. In an attempt to understand the overall

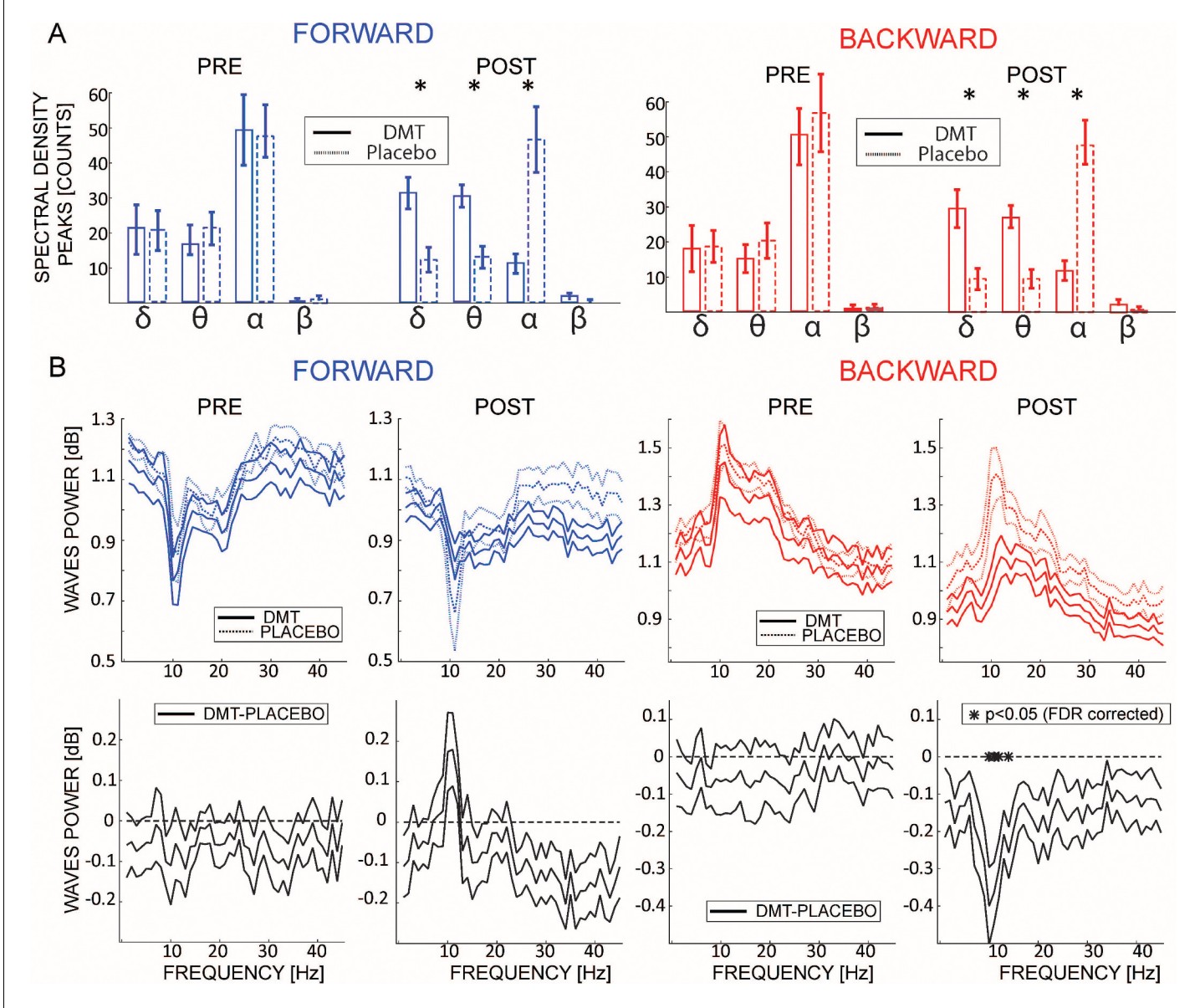

**Figure 3.** DMT influences the frequency of the travelling waves. (**A**) Left and right panels show the waves' frequencies computed from the maximum value from each quadrant in the 2D-FFT map for FW and BW waves, pre- and post-infusion. The histogram reflects the average between participants of the number of 1 s time-windows having a wave peak at the corresponding frequency. Notably, DMT significantly reduces α and β band oscillations, while enhancing δ and θ. Asterisks denote significant differences between DMT and Placebo conditions. (**B**) The upper panels show the amount of waves computed at each frequency of the 2D-FFT (i.e. not considering the maximum power per quadrant as in (**A**), but considering it for each frequency), for FW and BW waves, pre- and post-infusion. As shown in previous analysis, DMT induces an overall decrease of spectral power, especially in the alpha band BW waves, with the notable exception of an increase in FW waves in the alpha range.

relationship between FW and BW waves, we focused on the minutes when both BW and FW waves were significantly larger than 0 (minutes 2 to 5 after DMT injection, see *Figure 2A*). On these data we performed a moment-by-moment correlation between their respective net amount (as measured in decibel – see *Figure 1*). We found a clear and significant negative relationship (Bayesian t-test against zero, pre-DMT $BF_{10}$ = 393.1, error <0.0001%, 95% CI: [−0.448,−0.212]; Post-DMT $BF_{10}$ = 381.9, error <0.0001%, 95% CI: [−0.479,−0.226]), very consistent across participants and irrespective of DMT injection (difference between pre- and post-, Bayesian t-test $BF_{10}$ = 0.225; error<0.02% *Figure 4*, first panel). This result demonstrates that, in general, FW waves tend to be weaker whenever BW waves are stronger, and vice versa. In other words, FW and BW remain

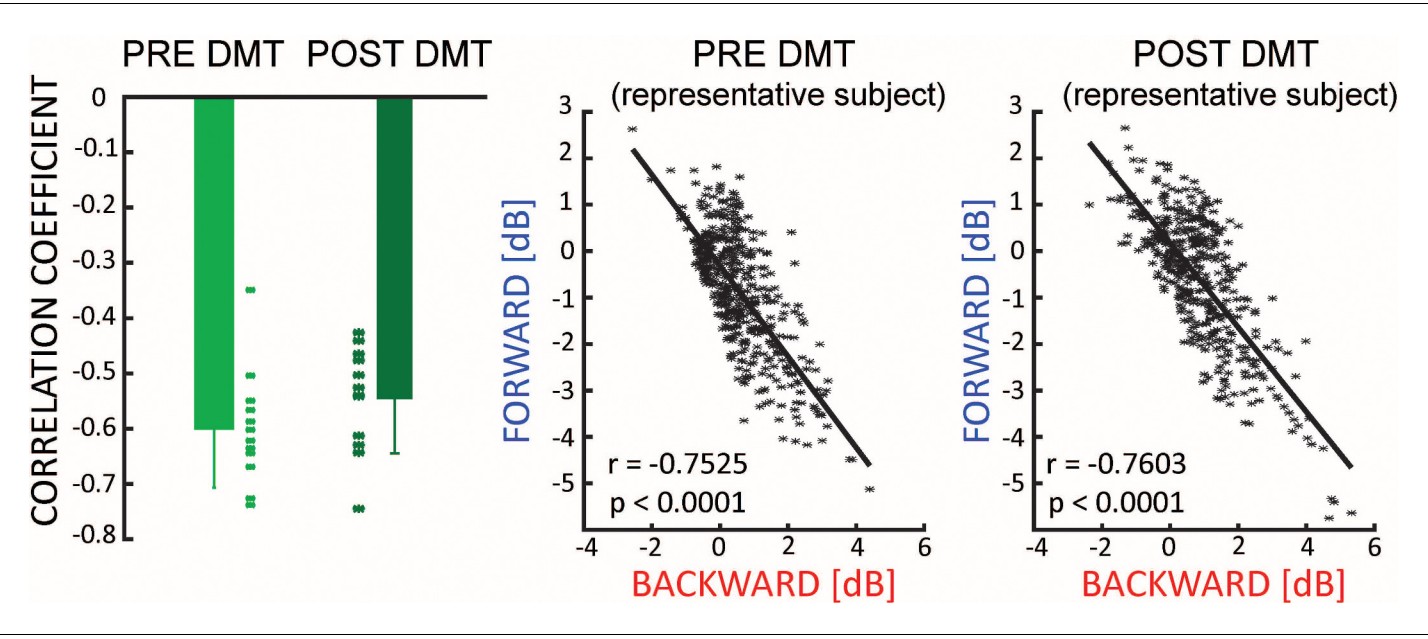

**Figure 4.** Travelling waves directions. There is a negative correlation between the net amount of FW and BW waves, which is not influenced by the ingestion of DMT (left panel). The middle and the right panel show the relationship for a typical subject pre- and post-DMT injection.

present after drug injection, sum to a consistent total amount, and remain inversely related; it is only the ratio of contribution from each that changes after DMT (i.e. less BW, more FW waves).

## Is there a correlation between waves and subjective reports?

We investigated whether changes in travelling waves under DMT correlated with the subjective effects of the drug. Specifically, for 20 min after DMT injection participants provided an intensity rating every minute and, when subjective effects faded, participants filled various questionnaires that addressed different aspects of the experience (see *Timmermann et al., 2019* for details). First, we found a robust correlation between minute-by-minute intensity rates and the amplitude of the waves, as shown in the first panel of *Figure 5*. This result reveals that the developing intensity of the drug's subjective effects and changes in the amplitude of waves correlate positively (FW) or negatively (BW) across time, both peaking a few minutes after drug injection. Second, treating each time point independently, we again correlated intensity ratings with the amount of each wave type, across subjects. The middle panel of *Figure 5* shows a clear trend for the correlation coefficients over time. Despite the limited number of data-points (n = 12), the correlation coefficients reach high values (~0.4), implying that, around the moment where the drug had its maximal effect (2–5 min after injection), those subjects who reported the most intense effects were also those who had the strongest travelling waves in the FW direction, and the weakest waves in the BW direction. Finally, we correlated the amount of FW and BW waves with ratings focused specifically on visual imagery: remarkably, ratings of all of the relevant questionnaire items correlated strongly with the increased amount of FW waves under DMT. As the same relationship was not apparent for the BW waves, this consolidates the view that visionary experiences under DMT correspond to higher amounts of FW waves in particular. Taken together with previous results from visual stimulation experiments independent of DMT (*Pang et al., 2020*), these data strongly support the principle that cortical travelling waves (and increased FW waves in particular) correlate with the conscious visual experiences, whether induced exogenously (via direct visual stimulation) or endogenously (visionary or hallucinatory experiences).

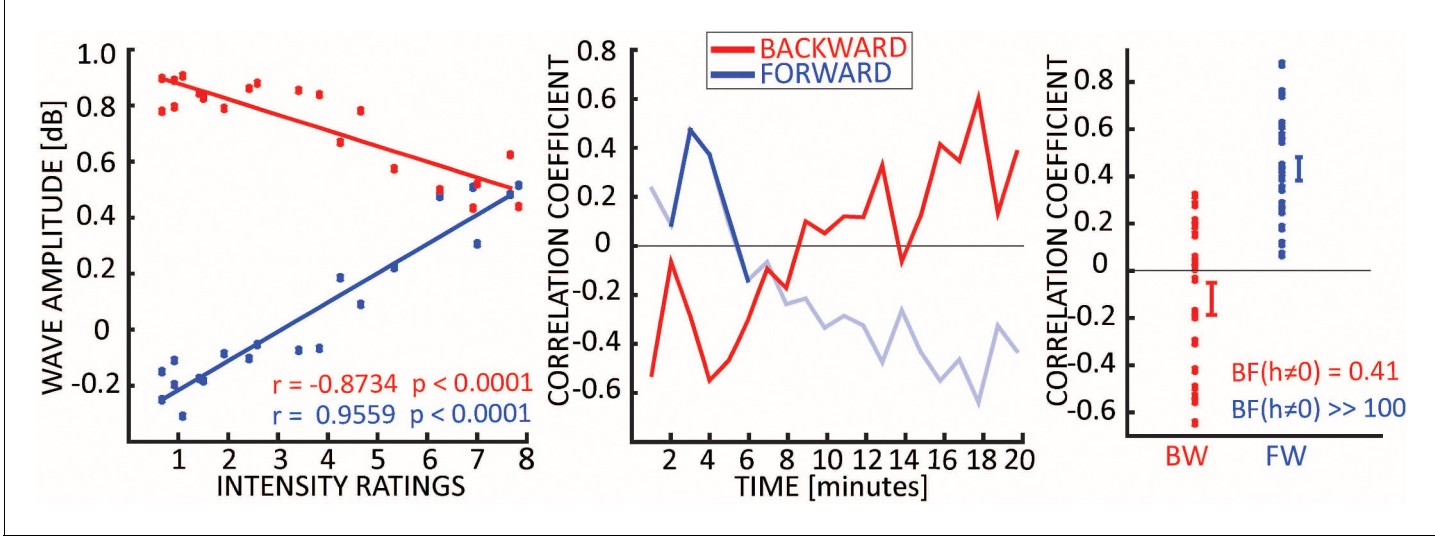

**Figure 5.** Travelling waves vs subjective ratings. The first panel shows the correlation between intensity rate and waves amplitude across time-points. Each dot represents a one-minute time-bin from DMT injection, the x-axis reflects the average intensity rating across subjects, and the y-axis indicates the average strength of BW or FW waves across subjects (both correlations p<0.0001). The middle panel shows the correlation coefficients across participants, obtained by correlating the intensity ratings and the waves' amount separately for each time point. Solid lines show when the amount of waves is significantly larger than zero (always for BW waves, few minutes after DMT injections for FW waves – see *Figure 2A*). However, given the limited statistical power (N = 12), and proper correction for multiple testing, correlations did not reach significance at any time point. The last panel shows the correlation coefficients between the visual imagery specific ratings provided at the end of the experiment (i.e. Visual Analogue Scale, see methods) and the net amount of waves (measured when both BW and FW were significantly different than zero, i.e. from minutes 2 to 5): for all 20 items in the questionnaire there was a positive trend between the amount of FW waves and the intensity of visual imagery, as confirmed by a Bayesian t-test against zero (BF for FW waves >> 100). We did not observe this effect in the BW waves (BF = 0.41).

## Discussion

In this study we investigated the effects of the classic serotonergic psychedelic drug DMT on cortical spatio-temporal dynamics typically described as travelling waves (*Muller et al., 2018*). We analysed EEG signals recorded from 13 participants who kept their eyes closed while receiving drug. Results revealed that, compared with consistent eyes-closed conditions under placebo, eyes-closed DMT is associated with striking changes in cortical dynamics, which are remarkably similar to those observed during actual eyes-open visual stimulation (*Alamia and VanRullen, 2019*; *Pang et al., 2020*). Specifically, we observed a reduction in BW waves, and increase in FW ones, as well as an overall decrease in α band (8–12 Hz) oscillatory frequencies (*Timmermann et al., 2019*). Moreover, increases in the amount of FW waves correlated positively with real-time ratings of the subjective intensity of the drug experience as well as post-hoc ratings of visual imagery, suggesting a clear relationship between travelling waves and a distinct and novel type of conscious experience.

### Relation to previous findings

Initiated by the discovery of mescaline, and catalysed by the discovery of LSD, Western medicine has explored the scientific value and therapeutic potential of psychedelic compounds for over a century (*Carhart-Harris, 2018*; *Schoen, 1964*; *Strassman, 1995b*). DMT has been evoking particular interest in recent decades, with new studies into its basic pharmacology (*Dean et al., 2019*), endogenous function (*Barker et al., 2012*) and effects on cortical activity in rats (*Artigas et al., 2016*; *Riga et al., 2014*) and humans (*Daumann et al., 2010*; *de Araujo et al., 2012*; *Valle et al., 2016*). There has been a surprising dearth of resting-state human neuroimaging studies involving pure DMT (*Palhano-Fontes et al., 2015*; *Timmermann et al., 2019*) which, given its profound and basic effects on conscious awareness, could be viewed as a scientific oversight.

Previous work involving ayahuasca and BOLD fMRI found increased visual cortex BOLD signal under the drug vs placebo while participants engaged in an eyes-closed imagery task – a result that

was interpreted as consistent with the 'visionary' effects of ayahuasca (*de Araujo et al., 2012*). Despite some initial debate (*Bartolomeo, 2008*), it is now generally accepted that occipital cortex becomes activated during visual imagery (*Fulford et al., 2018*; *Pearson, 2019*). Placing these findings into the context of previous work demonstrating increased FW travelling waves during direct visual perception (*Alamia and VanRullen, 2019*; *Pang et al., 2020*), our present findings of increased FW waves under DMT correlating with visionary experiences lend significant support to the notion that DMT/ayahuasca – and perhaps other psychedelics – engage the visual apparatus in a fashion that is consistent with actual exogenously driven visual perception. Future work could extend this principle to other apparently endogenous generated visionary experiences such as dream visions and other hallucinatory states. We would hypothesize a consistent favouring of FW waves during these states. If consistent mechanisms were also found to underpin hallucinatory experiences in other sensory modalities – such as the auditory one, a basic principle underlying sensory hallucinations might be established.

## Pharmacological considerations

As a classic serotonergic psychedelic drug, DMT's signature psychological effects are likely mediated by stimulation of the serotonin 2A (5-HT2A) receptor subtype. As with all other classic psychedelics (*Nichols, 2016*) the 5-HT2A receptor has been found to be essential for the full signature psychological and brain effects of Ayahuasca (*Valle et al., 2016*). In addition to its role in mediating altered perceptual experiences under psychedelics, the 5-HT2A receptor has also been linked to visual hallucinations in neurological disorders, with a 5-HT2A receptor inverse agonist having been licensed for hallucinations and delusions in Parkinson's disease with additional evidence for its efficacy in reducing consistent symptoms in Alzheimer's disease (*Ballard et al., 2018*). Until recently, a systems level mechanistic account of the role of 5-HT2A receptor agonism in visionary or hallucinatory experiences has, however, been lacking.

## Predictive coding and psychedelics

There is a wealth of evidence that Bayesian or predictive mechanisms play a fundamental role in cognitive and perceptual processing (*den Ouden et al., 2012*; *Kok and De Lange, 2015*) and our understanding of the functional architecture underlying such processing is continually being updated (*Alamia and VanRullen, 2019*; *Friston, 2018*). According to predictive coding (*Huang and Rao, 2011*), the brain strives to be a model of its environment. More specifically, based on the assumption that the cortex is a hierarchical system – message passing from higher cortical levels is proposed to encode predictions about the activity of lower levels. This mechanism is interrupted when predictions are contradicted by the lower-level activity ('prediction error') – in which case, information passes up the cortical hierarchy where it can update predictions. Predictive coding has recently served as a guiding framework for explaining the psychological and functional brain effects of psychedelic compounds (*Carhart-Harris and Friston, 2019*; *Pink-Hashkes et al., 2017*). According to one model (*Carhart-Harris and Friston, 2019*), psychedelics decrease the precision- weighting of top-down priors, thereby liberating bottom-up information flow. Various aspects of the multi-level action of psychedelics are consistent with this model, such as the induction of asynchronous neuronal discharge rates in cortical layer 5 (*Celada et al., 2008*), reduced alpha oscillations (*Carhart-Harris et al., 2016*; *Muthukumaraswamy et al., 2013*) increased signal complexity (*Schartner et al., 2017*; *Timmermann et al., 2019*) and the breakdown of large-scale intrinsic networks (*Carhart-Harris et al., 2016*).

Recent empirically supported modelling work has lent support to assumptions that top-down predictions and bottom-up prediction-errors are encoded in the direction of propagating cortical travelling waves (*Alamia and VanRullen, 2019*). Specifically, these simulations demonstrated that a minimal predictive coding model implementing biologically plausible constraints (i.e. temporal delays in the communication between regions and time constants) generates alpha-band travelling waves, which propagate from frontal to occipital regions and vice versa, depending on the 'cognitive states' of the model (input-driven vs. prior-driven), as confirmed by EEG data in healthy participants (in that case, processing visual stimuli vs. closed-eyes resting state).

The view that predictive coding could be the underlying principle explaining both the propagation of alpha-band travelling waves and the neural changes induced by psychedelics opened-up a

tantalizing opportunity for testing assumptions both about the nature of travelling waves and how they should be modulated by psychedelics (*Carhart-Harris and Friston, 2019*). Although we are restricted to speculation by the lack of direct experimental manipulation of top-down and bottom-up sensory inputs, our prior assumptions were so emphatically endorsed by the data, including how propagation-shifts related to subjective experience, that, in-line with prior hypotheses and motivations for the analyses, we were persuaded to infer about both the functional relevance of cortical travelling waves and brain action of psychedelics. Additional studies manipulating bottom-up and top-down analysis of sensory inputs with alternative perceptual designs will be required to confirm the relation between predictive coding, alpha-band oscillatory travelling waves and psychedelics states. Moreover, future studies can now be envisioned to examine how these assumptions translate to other phenomena such as non-drug induced visionary and hallucinatory states.

## Conclusion

The present analyses were applied to the first EEG data on the effects of DMT on human resting-state brain activity. In-line with a specific prior hypothesis, clear evidence was found of a shift in cortical travelling waves away from the normal basal predominance of backward waves and towards the predominance of forward waves – remarkably similar to what has been observed during eyes-open visual stimulation. Moreover, the increases in forward waves correlated positively with both the general intensity of DMT's subjective effects, as well as its more specific effects on eyes-closed visual imagery. These findings have specific and broad implications: for the brain mechanisms underlying the DMT/psychedelic state as well as conscious visual perception more fundamentally.

# Materials and methods

## Participants and experimental procedure

In this study we analysed a dataset presented in a previous publication (*Timmermann et al., 2019*), to address a very different scientific question using another analytical approach. Consequently, the information reported in this and the next paragraphs overlaps with the previous study (to which we refer the reader for additional details). Thirteen participants took part in this study (six females, age 34.4 ± 9.1 SD), sample size was chosen based on prior EEG and MEG studies and effect sizes with similar compounds. All participants provided written informed consent, and the study was approved by the National Research Ethics (NRES) Committee London – Brent and the Health Research Authority. The study was conducted in-line with the Declaration of Helsinki and the National Health Service Research Governance Framework.

Participants were carefully screened before joining the experiments. A medical doctor conducted physical examination, electrocardiogram, blood pressure and routine blood tests. A successful psychiatric interview was necessary to join the experiment. Other exclusion criteria were (1) under 18 years of age, (2) having no previous experience with psychedelic drugs, (3) history of diagnosed psychiatric illnesses, (4) excessive use of alcohol (more than 40 units per week). The day before the experiment a urine and pregnancy test (when applicable) were performed, together with a breathalyzer test.

Participants attended two sessions, in the first one, they received placebo, while DMT was administered in the second session. We employed a fixed-order, single blind design considering that psychedelics have been shown to induce lasting psychological changes (*Maclean et al., 2011*), which could have led to confounding effects on the following session if DMT had been administered in the first session. Additionally, we aimed at ensuring familiarity with the research environment and the study team before providing the psychedelics compound. Given the lack of human data with DMT, progressively increasing doses were provided to different participants (four different doses were used: 7, 14, 18 and 20 mg, to 3, 4, 1, and 5 successive participants, respectively). EEG signals were recorded before and up to 20 min after drug delivery. Participants rested in a semi-supine position with their eyes closed during the duration of the whole experiment. The eyes-closed instruction was confirmed by visual inspection of the participants during dosing. At each minute, participants provided an intensity rating, while blood samples were taken at given time-points (the same for placebo and DMT conditions) via a cannula inserted in the participants' arm. One day after the DMT session, participants reported their subjective experience completing an interview composed of several

questionnaires (see *Timmermann et al., 2019* for details). In this study we focused on the Visual Analogue Scale values.

## EEG preprocessing

EEG signals were recorded using a 32-channels Brainproduct EEG system sampling at 1000 Hz. A high-pass filter at 0.1 Hz and an anti-aliasing low-pass filter at 450 Hz were applied before applying a band-pass filter at 1–45 Hz. Epochs with artifacts were manually removed upon visual inspection. Independent-component analysis (ICA) was performed and components corresponding to eye-movement and cardiac-related artifacts were removed from the EEG signal. The data were re-referenced to the average of all electrodes. All the preprocessing was carried out using the Fieldtrip toolbox (*Oostenveld et al., 2011*), while the following analysis were run using custom scripts in MATLAB.

## Waves quantification

We epoched the preprocessed EEG signals in 1 s windows, sliding with a step of 500 ms (see *Figure 1*). For each time-window, we then arranged a 2D time-electrode map composed of five electrodes (i.e. Oz, POz, Pz, Cz, FCz). From each map we computed the 2D Fast Fourier Transform (2DFFT – *Figure 1*), from which we extracted the maximum value in the upper and lower quadrants, representing respectively the power of forward (FW) and backward (BW) waves. We also performed the same procedure 100 times after having randomised the electrodes' order: the surrogate 2D-FFT spectrum has the same temporal frequency content overall, but the spatial information is disrupted, and the information about the wave directionality is lost. In such a manner we obtained the null or surrogate measures, namely FWss and BWss, whose values are the average over the 100 repetitions (see *Figure 1*). Eventually, we computed the actual amount of waves in decibel (dB), considering the log-ratio between the actual and the surrogate values:

$$\text{FWdb} = 10*\log_{10}\frac{\text{FW}}{\text{FWss}} \qquad \text{BWdb} = 10*\log_{10}\frac{\text{BW}}{\text{BWss}}$$

It is worth noting that this value represents the amount of significant waves against the null distribution, that is against the hypothesis of having no FW or BW waves.

## Statistical analysis

All the analyses regarding the EEG signals were performed in MATLAB. Bayesian analyses were run in JASP (*Team, 2018*). We ran separate Bayesian ANOVA for FW and BW conditions, and we considered as factors the time of injection (pre-post, see *Figure 2A*) and drug condition (DMT vs Placebo). Subjects were considered to account for random factors. Regarding the minute-by-minute analysis (*Figure 2A*, right panels), we performed a t-test at each time-bin against zero, and we corrected all the p-values according to the False Discovery Rate (*Benjamini and Yekutieli, 2005*). All data and code to perform the analysis are available at https://osf.io/wujgp/.

# Acknowledgements

We dedicate this paper to the memory of *Jordi Riba*, a gracious man and pioneering psychedelic researcher.

# Additional information

### Funding

| Funder | Grant reference number | Author |
|---|---|---|
| Alexander Mosley Charitable Trust | | Robin L Carhart-Harris |
| Ad Astra Chandaria Foundation | | Robin L Carhart-Harris |
| CRCNS ANR-NSF | ANR-19-NEUC-0004 | Rufin VanRullen |

| ANITI (Artificial and Natural Intelligence Toulouse Institute) Research Chair | ANR-19-PI3A-0004 | Rufin VanRullen |
| Comision Nacional de Investigacion Cientifica y Tecnologica de Chile | | Christopher Timmermann |

The funders had no role in study design, data collection and interpretation, or the decision to submit the work for publication.

### Author contributions

Andrea Alamia, Software, Formal analysis, Visualization, Methodology, Writing - original draft; Christopher Timmermann, Conceptualization, Data curation, Formal analysis, Methodology, Writing - review and editing; David J Nutt, Conceptualization, Data curation, Supervision, Funding acquisition; Rufin VanRullen, Formal analysis, Supervision, Funding acquisition, Methodology, Writing - review and editing; Robin L Carhart-Harris, Conceptualization, Data curation, Supervision, Funding acquisition, Project administration, Writing - review and editing

### Author ORCIDs

Andrea Alamia (iD) https://orcid.org/0000-0001-9826-2161
Christopher Timmermann (iD) https://orcid.org/0000-0002-2281-377X
Rufin VanRullen (iD) https://orcid.org/0000-0002-3611-7716

### Ethics

Human subjects: All participants provided written informed consent, and the study was approved by the National Research Ethics (NRES) Committee London - Brent and the Health Research Authority (16/LO/0897). The study was conducted in line with the Declaration of Helsinki and the National Health Service Research Governance Framework.

### Decision letter and Author response

Decision letter https://doi.org/10.7554/eLife.59784.sa1
Author response https://doi.org/10.7554/eLife.59784.sa2

## Additional files

### Supplementary files

• Transparent reporting form

### Data availability

The data and the code to perform the analysis are available at : https://osf.io/wujgp/.

The following dataset was generated:

| Author(s) | Year | Dataset title | Dataset URL | Database and Identifier |
| --- | --- | --- | --- | --- |
| Alamia A | 2020 | DMT alters cortical travelling waves | https://osf.io/wujgp/ | Open Science Framework, 10.17605/OSF.IO/WUJGP |

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
