## [Decision Letter]

**Acceptance summary:**

In this study, Alamia and colleagues describe the effect of N,N, Dimethyltryptamine DMT on the resting-state dynamics of α traveling waves. DMT is a serotonergic psychedelic drug that elicits vivid hallucinations. The authors tested whether DMT provoke a relative change in strength of forward- and backward- α traveling waves recorded with non-invasive electroencephalography. Specifically, the hypothesis was that traveling waves under DMT may show features of visual perception in the absence of photic stimulation. Indeed, following DMT consumption and during eyes-closed resting state, the authors report an increase of forward-traveling waves, an α power increase (comparable to photic stimulation) and of low-frequency components in the low-range spectrum. The implication of traveling waves are discussed in relation to predictive coding.

**Decision letter after peer review:**

Thank you for submitting your article "DMT alters cortical travelling waves" for consideration by *eLife*. Your article has been reviewed by three peer reviewers, one of whom is a member of our Board of Reviewing Editors, and the evaluation has been overseen by Timothy Behrens as the Senior Editor. The following individual involved in review of your submission has agreed to reveal their identity: David Murray Alexander (Reviewer #2).

The reviewers have discussed the reviews with one another and the Reviewing Editor has drafted this decision to help you prepare a revised submission.

Summary:

Alamia and colleagues describe the effect of N,N, Dimethyltryptamine DMT on human resting-state dynamics recorded with non-invasive EEG. DMT is a serotonergic psychedelic drug that elicits vivid hallucinations. The authors propose that DMT provokes a relative change in strength of forward- and backward- α [~10 Hz] traveling waves, indicative of bottom-up and top-down propagations of information. In three main analyses of EEG recordings following the administration of DMT, the authors report an increase of forward-traveling waves, α power (comparable to photic stimulation) and low-frequency components in the low-range spectrum.

I highlight three main issues that need to be addressed in a revised paper.

1) All three reviewers highlight specific points re. the limitations of the current analysis (e.g. a prior choice of electrodes) and the choice for quantifications of the traveling waves. The authors should clarify their rationale underlying their analytical choices. As suggested by reviewer 3, a section dedicated to "Quantifying travelling waves" may be helpful.

2) All three reviewers raised substantial concerns about the interpretability of scalp level recordings in relation to the generators of the signals as well as the inference that can be made on the traveling pattern. Reviewer 1 raised concerns about the full spectral changes that can be seen and reviewer 3 suggested the possibility of interference patterns.

3) Reviewers 1 and 2 consider the predictive coding hypothesis a far-stretched inference of current results, and thus needs to be refined.

Reviewer #1:

In this study, Alamia and colleagues describe the effect of N,N, Dimethyltryptamine DMT on the resting-state dynamics of α traveling waves. DMT is a serotonergic psychedelic drug that elicits vivid hallucinations. The authors herein propose that DMT provokes a relative change in strength of forward- and backward- α traveling waves recorded with non-invasive EEG. The authors hypothesized that traveling waves under DMT may show features of visual perception, namely an increase of forward going (occipital to frontal) traveling waves during eyes-closed resting-state thus independently of photic inputs. With three main analyses, the authors observe that following DMT, forward-traveling waves increase, α power increase (comparable to photic stimulation) and an overall increase of low-frequency components in the low-range spectrum.

I have several major concerns regarding the reliability of the analysis and subsequent interpretations. One is that while the authors quantified traveling waves, they also report a radical change in the overall spectral fingerprinting of the EEG (Figure 3) and clearly showing that α is largely suppressed following DMT thus largely diminishing the reliability and pertinence of focusing on α traveling wave while low-frequency are largely boosted. Second, the authors report consistent inverse relationships between FW and BW waves, which may simply result from moving dipoles that generate the signals; in light of this, the inverse relation in Figure 5 between FW/BW is not surprising. Finally, would a simpler measure of decrease in α power and increase of low spectral power reveal similar correlations to behavior?

Reviewer #2:

The authors present a sensor level analysis of traveling waves in the EEG, during dosage with DMT or saline. The work is a strong contribution to the study of cortical traveling waves (TWs) due to the pharmacological manipulation, which helps our understanding of the causal role of TWs. This contribution is bolstered by being able to draw parallels with the effects of visual stimulation (or not) during rest, reported in a contemporaneous manuscript. The manuscript will be of general interest to readers of *eLife*. The manuscript is crisply written.

The conclusions follow from the analysis.

Concerns:

1) The quantitative methods to analyse TWs are rather week, being focussed on direction of travel along the anterior-posterior axis. While these methods are sufficient to support the main conclusions, more could be teased from the experiment by following recent developments in TW quantification.

For example, using peak tracing along the lines of Massimini et al., 2004, would enable the detailed paths of the waves to be traced over the whole recording array, and velocity to be estimated.

Methods exist to estimate the spatial frequency of the waves on the scalp, as well as the proportion of traveling vs. standing waves (Alexander et al., 2016). Likewise, other directional components of the wave trajectory can be assessed by using PCA to create a spatial basis for the waves (Alexander et al., 2006, 2009, 2013).

It seems possible that important features of the data have been missed by limiting the analysis to electrodes FCz to Oz. For example, what if DMT influence and visual stimulation share a common primary direction, as is found, but DMT waves are more left posterior to right anterior and visual stimulation is more right posterior to left anterior (or vice versa)?

2) The sections on predictive coding are only tenuously supported by the data. In particular, I can see no discussion on how directional differences in the α band may be significant in this regard. What about situations where anterior-posterior differences are found in the δ band (Alexander et al., 2006; 2009)? Or if directional results were in another band? Because of the lack of specificity to this discussion, and the lack of explicit tests of this theoretical framework, I suggest these concepts be given a more appropriate weighting (less).

3) An obvious objection to the analysis is that it is sensor level. The authors need to address their reasons for doing this e.g. that source projections destroy real long-range correlations as well as blurring by the scalp and other tissues. See Nunez, 1974; Nunez et al., 1997; Freeman et al., 2000; 2003 and Alexander et al., 2019, for a summary of these issues.

Reviewer #3:

This study of the effects of the drug DMT on the direction and occurrence of EEG traveling waves seems generally plausible to me, although some important aspects are not discussed. I don't have major criticisms. However, as one who has studied EEG traveling and standing waves for many years, I worry that those unfamiliar with EEG wave phenomena may misinterpret some of these results given their partial dependence on the specific experimental methods employed. While I have not read previous papers by these authors that may fill in some of the important gaps, I list below some ideas that any reader interested in EEG waves and their neuro-scientific interpretation must be aware of. A summary paragraph in the section "Quantifying travelling waves" is recommended concerning the following basic concepts that must be understood if the results are to be interpreted correctly.

1) In all but the simplest systems, traveling waves occur in groups (packets) over some range of spatial wavelengths (multiple spatial frequencies, k). This is to be expected in brains, based on both theory and experiment (see Nunez and Srinivasan, 2006; 2014; Nunez, 2000).

2) Any experimental electrode array will be sensitive to only parts of these wave packets, e.g. only waves shorter than the spatial extent of the array and waves longer than twice the electrode separation distance (Nyquist criterion in space) can be resolved. In scalp recordings, the shorter waves may be mostly removed by volume conduction.

3) As a consequence of #2, waves recorded directly from the cortex (as indicated in several recent studies) will emphasize shorter waves than the scalp recorded waves. In the case of small cortical arrays, the ECoG overlap with scalp data may be minimal. Thus, the different estimated wave properties (including propagation direction) need not agree.

4) When waves are traveling in multiple directions at nearly the same time in "closed" systems (e.g., the cortical/white matter), there are only two possible results. Either the waves must damp out or they combine (interfere) to form standing waves (e.g. α waves traveling both forward and backward). One expects that the actual behavior depends on brain state, including the influence of drugs (see Nunez and Srinivasan, 2006; 2014; Nunez, 2000).

---

## [Author Response]

Revisions for this paper:I highlight three main issues that need to be addressed in a revised paper.1) All three reviewers highlight specific points re. the limitations of the current analysis (e.g. a prior choice of electrodes) and the choice for quantifications of the traveling waves. The authors should clarify their rationale underlying their analytical choices. As suggested by reviewer 3, a section dedicated to "Quantifying travelling waves" may be helpful.

We have now acknowledged the limitations of the current analyses, and we performed several additional steps to improve on our previous approach. As explained in detail in what follows (and in the revised manuscript) we included a new analysis considering different lines of electrodes, spanning from posterior to anterior, but also left to right brain regions. We also performed an additional complementary analysis based on the suggestions of reviewer 2, showing similar results as our original analysis. Finally, as suggested by reviewer 3, we integrated the current “Quantifying travelling waves” paragraph with his generous suggestions, improving the readability of the manuscript to a non-specialized audience. We believe that these modifications will satisfy both the editor and all the reviewers.

2) All three reviewers raised substantial concerns about the interpretability of scalp level recordings in relation to the generators of the signals as well as the inference that can be made on the traveling pattern. Reviewer 1 raised concerns about the full spectral changes that can be seen and reviewer 3 suggested the possibility of interference patterns.

We agree with the concerns raised by both reviewers, and we have included an additional analysis (now Figure 3B in the revised manuscript) that addresses specifically these concerns. Regarding the spectral changes, our new analysis avoids choosing *a priori* a single frequency band (i.e. the one corresponding to the global maximum in the 2D-FFT) but instead analyze the changes in all the spectrum. This novel approach, besides confirming our previous results, provides a fuller view of the overall changes in the spectral pattern induced by DMT. Concerning the source generators of the travelling waves pattern, we discuss this point in the revised manuscript, arguing that a sensor analysis is more appropriate in this case because it circumvents some limitations related specifically to source analysis (e.g. source projections impair long-range connections). Finally, as shown in Figure 4, backward and forward waves were negatively correlated on a trial by trial basis, which will tend to limit the possibility suggested by reviewer 3 of having interference patterns (resulting in standing waves).

3) Reviewers 1 and 2 consider the predictive coding hypothesis a far-stretched inference of current results, and thus needs to be refined.

We addressed carefully this point by giving overall less weight to the Predictive Coding hypothesis in the Discussion, as suggested by both reviewers. Additionally, as suggested specifically by reviewer 2, we clarify in the revised Discussion the link between Predictive Coding, α oscillations and travelling waves, and the motivation behind our original hypothesis and the relationship with the present results. More precisely, we previously demonstrated how a model based on Predictive Coding principles and implementing biologically plausible constraints gives rise to α-band travelling waves, whose direction of propagation depends on the “cognitive” state of the model/subject (FW during visual stimulation, BW during closed-eyes resting state). Then, starting from the premise that psychedelics disrupt prior distributions encoded in hierarchically high-level properties of brain function (Carhart-Harris and Friston, 2019), we formulated the specific hypothesis that DMT could specifically disrupt α-band travelling waves, enhancing their feed-forward propagation while decreasing the feed-back direction. All in all, we found it remarkable that such a specific hypothesis received such clear support in the data. However, we understand that interpretations can always be queried and more work is needed to scrutinize the one we offer based on our specific prior hypothesis. We have now rephrased our interpretation substantially in the revised version of the manuscript, to soften our conclusions and emphasize the need for more research.

Reviewer #1:In this study, Alamia and colleagues describe the effect of N,N, Dimethyltryptamine DMT on the resting-state dynamics of α traveling waves. DMT is a serotonergic psychedelic drug that elicits vivid hallucinations. The authors herein propose that DMT provokes a relative change in strength of forward- and backward- α traveling waves recorded with non-invasive EEG. The authors hypothesized that traveling waves under DMT may show features of visual perception, namely an increase of forward going (occipital to frontal) traveling waves during eyes-closed resting-state thus independently of photic inputs. With three main analyses, the authors observe that following DMT, forward-traveling waves increase, α power increase (comparable to photic stimulation) and an overall increase of low-frequency components in the low-range spectrum.I have several major concerns regarding the reliability of the analysis and subsequent interpretations. One is that while the authors quantified traveling waves, they also report a radical change in the overall spectral fingerprinting of the EEG (Figure 3) and clearly showing that α is largely suppressed following DMT thus largely diminishing the reliability and pertinence of focusing on α traveling wave while low-frequency are largely boosted. Second, the authors report consistent inverse relationships between FW and BW waves, which may simply result from moving dipoles that generate the signals; in light of this, the inverse relation in Figure 5 between FW/BW is not surprising. Finally, would a simpler measure of decrease in α power and increase of low spectral power reveal similar correlations to behavior?

We thank the reviewer for raising these important considerations. Regarding the relationship between the spectral changes in the EEG and the changes in the amount of waves, we performed an additional analysis quantifying these changes as a function of each frequency (i.e. extracting in the 2D-FFT the maximum value separately for each frequency). The first row of Author response image 1 (integrated in the revised version of the manuscript as figure 3B) shows the amount of FW and BW waves (in dB –as compared to the surrogate distribution) before and after DMT or Placebo infusion (i.e. Pre and Post). The second row shows the difference in the amount of waves between DMT and Placebo for each frequency. Interestingly the largest changes occur in the α band frequency for both forward and backward waves, even though after correcting for multiple comparison we found a significant reduction only in the BW α band. This analysis shows that, as suggested by the reviewer, the changes in the spectral fingerprint of the EEG do influence the waves’ propagation in several frequencies, but the largest changes systematically occur in the α band. This additional analysis has been introduced in the Results section (paragraph: “Does DMT influence the frequency of travelling waves?”)

**Author response image 1. sa2fig1:** The upper panels show the amount of waves computed at each frequency of the 2D-FFT (i.e. not considering the maximum power per quadrant as in A, but considering it separately for each frequency), for FW and BW waves, pre- and post- infusion. The lower panels show the difference between DMT- and placebo-induced waves for each condition. As shown in the previous analysis, DMT induces an overall decrease of the waves’ amplitude, especially pronounced (and significant) in the α band BW waves, with the notable exception of FW waves in the α range, where an DMT-induced increase is observed.

Regarding the relationship between FW and BW waves, our analysis –shown in Figure 4 of the manuscript- suggests that after DMT –or during photic visual stimulation- FW and BW waves do not co-occur simultaneously, but tend to alternate temporally, as revealed by the negative correlation. We agree with the reviewer that moving dipoles could be responsible for the generation of these signals, as reported in an in-depth analysis of a previous paper investigating the source localization of similar waves patterns (Lozano-Soldevilla and VanRullen, 2019). In line with a similar comment from reviewer 2, we added in the manuscript a reference to source vs sensors analysis (“In addition, it is important to note that our waves’ analysis focuses on the sensor level, as source projections present a number of important limitations, such as impairing long-range connections, as well as smearing of signals due to scalp interference (Alexander et al., 2019; Freeman and Barrie, 2000; Nunez, 1974)”). Lastly, a previous analysis on the same dataset (Timmermann et al., 2019) identified a correlation between theta- and δ-band spectral power changes and subjective behavior, but not for changes in the α range. This suggests that the correlation reported in Figure 5 is not a direct consequence of changes in the EEG spectral power.

Reviewer #2:The authors present a sensor level analysis of traveling waves in the EEG, during dosage with DMT or saline. The work is a strong contribution to the study of cortical traveling waves (TWs) due to the pharmacological manipulation, which helps our understanding of the causal role of TWs. This contribution is bolstered by being able to draw parallels with the effects of visual stimulation (or not) during rest, reported in a contemporaneous manuscript. The manuscript will be of general interest to readers of eLife. The manuscript is crisply written.The conclusions follow from the analysis.Concerns:1) The quantitative methods to analyse TWs are rather week, being focussed on direction of travel along the anterior-posterior axis. While these methods are sufficient to support the main conclusions, more could be teased from the experiment by following recent developments in TW quantification.For example, using peak tracing along the lines of Massimini et al., 2004, would enable the detailed paths of the waves to be traced over the whole recording array, and velocity to be estimated.Methods exist to estimate the spatial frequency of the waves on the scalp, as well as the proportion of traveling vs. standing waves (Alexander et al., 2016). Likewise, other directional components of the wave trajectory can be assessed by using PCA to create a spatial basis for the waves (Alexander et al., 2006, 2009, 2013).

We thank the reviewer for his useful suggestions. As correctly noticed, our current method to detect travelling waves focuses exclusively on the Anterior-Posterior axis, in line with our previous studies (Alamia and VanRullen, 2019; Lozano-Soldevilla and VanRullen, 2019; Pang et al., 2020). However, we agree that more can be inferred from the data from other electrodes (see next point and comment to reviewer 1 for waves’ quantification on other axes). We applied a method similar to Alexander et al., 2006, 2009 and 2016, thus considering the phase of the signal over the entire array of electrodes. Specifically, we computed the phase of the α band-pass filtered signals pre- and post- DMT infusion, and referenced it to the central electrode Cz. The relative phase thus describes the propagation of the wave as compared to this electrodes: positive lags (in yellow in the Author response image 2) characterize earlier components, whereas negative lags (in blue) are associated with signals lagging behind. Author response image 2 summarizes the results in all conditions.

**Author response image 2. sa2fig2:** Relative phase obtained by computing the difference between the phases of the α band-pass filtered signal of each channel and Cz. Reassuringly, the pattern of results confirms the disruption of the top-down flow, counterbalanced by a bottom-up component, specifically after the infusion of DMT, in line with our original analysis.

Interestingly, we observed that after placebo, the typical top-down propagation of α-band waves remains unaltered, whereas after DMT, waves propagate both FW and BW, as revealed by an overall phase distribution around zero. Overall these results confirmed the one obtained with the 2D-FFT approach. We opted for keeping the latter for consistency with our previous studies (but we mentioned this result in the revised manuscript along with the references)

“Besides, in line with previous work on travelling waves (Alexander et al., 2013, 2006), an additional analysis based on relative phases of the α band-pass signals over all channels confirmed the same results, with DMT disrupting the typical top-down propagation of α-band waves (not shown).”

We agree with the reviewer that the approach used by (Massimini et al., 2004), would allow to identify the detailed path of the waves, and potentially their velocity. However, in their work, Massimini and colleagues targeted slow 1Hz waves (the signal was actually low-pass filtered at 4Hz); for each cycle waves were tracked based on the localization of the main (negative) peak whose voltage was below a threshold of -80 V. This approach, which provides reliable results for low-frequency waves, may present some non-trivial additional limitations when applied to higher frequencies. Specifically, the identification of each peak/cycle may not be straightforward for higher frequencies (e.g. α band); in Massimini et al. the time window used to identify the peak spanned between +/-800ms to the earliest peak, but such a window should be proportionally much shorter to isolate single peaks in the α range, and thus be increasingly more sensitive to jittered noise. We therefore favored the 2D-FFT approach, which –despite its own limitations- seemed more suitable to describe waves with higher temporal frequencies. Finally, regarding the waves speed, it is possible to estimate their velocity from the 2D-FFT, considering both spatial and temporal frequencies as shown in our previous study (Alamia and VanRullen, 2019). The reported results are consistent with the speed recorded for cortical waves (macroscopic scale, speed ~1.5 – 2.0 m/s (Muller et al., 2018)).

It seems possible that important features of the data have been missed by limiting the analysis to electrodes FCz to Oz. For example, what if DMT influence and visual stimulation share a common primary direction, as is found, but DMT waves are more left posterior to right anterior and visual stimulation is more right posterior to left anterior (or vice versa)?

We agree with the reviewer that the choice of the midline electrodes supports the main conclusion but prevents a broader view on the waves’ dynamic at the cortical level. Accordingly, and in line with the concerns of reviewer 1, we explored different lines of electrodes, to identify other axes of propagation. As shown in Figure 2—figure supplement 2, comparing PRE and POST DMT infusion reveals an increase of waves propagating from posterior to anterior regions considering an array of electrodes arranged from right posterior to left anterior (diag1 in Figure 2—figure supplement 2, Bayesian t-test BF=4.059, error=0.002%) and from left posterior to right anterior (diag2 in Figure 2—figure supplement 2, Bayesian t-test BF=4.848, error=0.0001%), similarly to the results obtained considering the main posterior-anterior axis (Bayesian t-test BF=5.4, error=0.001%). Additionally, we revealed a significant amount of waves (larger than 0 dB) propagating along the coronal line of electrodes (i.e. leftward and rightward), but those waves were not influenced by DMT infusion (for both leftward and rightward waves BF<0.4, error~0.02%). We included these analyses in the Results section and as Figure 2—figure supplement 2.

“In order to explore different propagation axes than the midline, we ran the same analysis on one array of electrodes running from posterior right to anterior left regions, and one from posterior left to anterior right ones: in both cases we obtained similar results as for the midline electrodes, that is, an increase and a decrease of FW and BW waves respectively following DMT infusion (see Figure 2—figure supplement 2).This suggests that the waves’ propagation spread to most posterior and frontal recording channels As a control, we additionally demonstrated that waves propagating from leftward to rightward regions (and vice versa) were not affected by DMT, as predicted by our hypothesis (see Figure 2—figure supplement 2).”

2) The sections on predictive coding are only tenuously supported by the data. In particular, I can see no discussion on how directional differences in the α band may be significant in this regard. What about situations where anterior-posterior differences are found in the δ band (Alexander et al., 2006; 2009)? Or if directional results were in another band? Because of the lack of specificity to this discussion, and the lack of explicit tests of this theoretical framework, I suggest these concepts be given a more appropriate weighting (less).

We agree and thank the reviewer for pointing out this shortcoming in the Discussion. The focus on the α-band originates from our previous study (Alamia and VanRullen, 2019) in which we demonstrated how a minimal Predictive Coding model implementing biologically plausible constraints (i.e. temporal delays in the communication between regions and time constants) generates α-band travelling waves whose direction of propagation is matched by experimental data. This result was the starting hypothesis that motivated the investigation of α-band travelling waves after DMT-infusion, under the hypothesis of the REBUS model (psychedelics disrupt prior distributions in higher brain regions- Carhart-Harris and Friston, 2019). In the revised version of the manuscript, we clarify the link between Predictive Coding, α oscillations and travelling waves. However, we agree with this reviewer (and reviewer 1 and the editor) that the Predictive Coding interpretation may not be directly but only indirectly supported by the current data, and so we have rephrased the relevant section and substantially softened it in the revised version of the manuscript, in accordance with this valid point.

“Specifically, these simulations demonstrated that a minimal Predictive Coding model implementing biologically plausible constraints (i.e. temporal delays in the communication between regions and time constants) generates α-band travelling waves, which propagate from frontal to occipital regions and vice versa, depending on the “cognitive states” of the model (input-driven vs prior-driven), as confirmed by EEG data in healthy participants (in that case, processing visual stimuli vs. closed-eyes resting state). The view that Predictive Coding could be the underlying principle explaining both the propagation of α-band travelling waves, and the neural changes induced by psychedelics opened-up a tantalizing opportunity for testing assumptions both about the nature of travelling waves and how they should be modulated by psychedelics (Carhart-Harris and Friston, 2019). Although we are restricted to speculation by the lack of direct experimental manipulation of top-down and bottom-up sensory inputs, our prior assumptions were so emphatically endorsed by the data, including how propagation-shifts related to subjective experience, that, in-line with prior hypotheses and motivations for the analyses, we were persuaded to infer about both the functional relevance of cortical travelling waves and brain action of psychedelics. Additional studies manipulating bottom-up and top-down analysis of sensory inputs with alternative perceptual designs will be required in order to confirm the relation between Predictive Coding, α-band oscillatory travelling waves and psychedelics states. Moreover, future studies can now be envisioned to examine how these assumptions translate to other phenomena such as non-drug induced visionary and hallucinatory states.”

3) An obvious objection to the analysis is that it is sensor level. The authors need to address their reasons for doing this e.g. that source projections destroy real long-range correlations as well as blurring by the scalp and other tissues. See Nunez, 1974; Nunez et al., 1997; Freeman et al., 2000; 2003 and Alexander et al., 2019, for a summary of these issues.

We thank the reviewer for the useful reminder. We included in the “Quantifying the waves” paragraph of the revised manuscript a few sentence addressing this issue:

“In addition, it’s important to note that our waves’ analysis focuses at the sensor level, as source projections presents few important limitations such as impairing long-range connections, as well as smearing the signals due to the scalp inference (Alexander et al., 2019; Freeman and Barrie, 2000; Nunez, 1974)”

Reviewer #3:This study of the effects of the drug DMT on the direction and occurrence of EEG traveling waves seems generally plausible to me, although some important aspects are not discussed. I don't have major criticisms. However, as one who has studied EEG traveling and standing waves for many years, I worry that those unfamiliar with EEG wave phenomena may misinterpret some of these results given their partial dependence on the specific experimental methods employed. While I have not read previous papers by these authors that may fill in some of the important gaps, I list below some ideas that any reader interested in EEG waves and their neuro-scientific interpretation must be aware of. A summary paragraph in the section "Quantifying travelling waves" is recommended concerning the following basic concepts that must be understood if the results are to be interpreted correctly.1) In all but the simplest systems, traveling waves occur in groups (packets) over some range of spatial wavelengths (multiple spatial frequencies, k). This is to be expected in brains, based on both theory and experiment (see Nunez and Srinivasan, 2006; 2014; Nunez, 2000).2) Any experimental electrode array will be sensitive to only parts of these wave packets, e.g. only waves shorter than the spatial extent of the array and waves longer than twice the electrode separation distance (Nyquist criterion in space) can be resolved. In scalp recordings, the shorter waves may be mostly removed by volume conduction.3) As a consequence of #2, waves recorded directly from the cortex (as indicated in several recent studies) will emphasize shorter waves than the scalp recorded waves. In the case of small cortical arrays, the ECoG overlap with scalp data may be minimal. Thus, the different estimated wave properties (including propagation direction) need not agree.4) When waves are traveling in multiple directions at nearly the same time in "closed" systems (e.g., the cortical/white matter), there are only two possible results. Either the waves must damp out or they combine (interfere) to form standing waves (e.g. α waves traveling both forward and backward). One expects that the actual behavior depends on brain state, including the influence of drugs (see Nunez and Srinivasan, 2006; 2014; Nunez, 2000).

We are very grateful to the reviewer for her/his overall positive opinion on our work, and her/his useful suggestions. As recommended, we integrated in the “Quantifying travelling waves” paragraph all the points listed above, with the corresponding references. We believe such changes improved the readability of the paper for those unfamiliar with EEG analysis, and hopefully will be satisfying and adequate for the reviewer.

“As demonstrated by both theoretical and experimental evidence (Nunez, 2000; Nunez and Srinivasan, 2014, 2009), in most systems, including the human brain, traveling waves occur in groups (or packets) over some range of spatial wavelengths having multiple spatial and temporal frequencies. Given any configurations of electrodes, only parts of these packets can be successfully detected, i.e. waves shorter than the spatial extent of the array, and waves longer than twice the electrode separation distance (Nyquist criterion in space). In scalp recordings, the shorter waves may be mostly removed by volume conduction. As a consequence, waves recorded directly from the cortex emphasize shorter waves than the scalp recorded waves. Specifically, in the case of small cortical arrays, the overlap between cortical and scalp data may be minimal, and the estimated wave properties (including propagation direction) may differ. Additionally, it is important to consider that when waves are traveling in multiple directions at nearly the same time in "closed" systems (e.g., the cortical/white matter), waves either damp out or interfere with each other to form standing waves (e.g. α waves traveling both forward and backward). It is reasonable to assume that the behavior of these properties will relate to global brain and mind states, and be sensitive to state-altering psychoactive drugs (Nunez, 2000; Nunez and Srinivasan, 2014, 2009).”

Reference:

Massimini M, Huber R, Ferrarelli F, Hill S, Tononi G. 2004. The sleep slow oscillation as a traveling wave. *J Neurosci***24**:6862–6870. doi:10.1523/JNEUROSCI.1318-04.2004